# Photooxidative stress-inducible orange and pink water-soluble astaxanthin-binding proteins in eukaryotic microalga

Shinji Kawasaki[1,2✉], Keita Yamazaki[2], Tohya Nishikata[2], Taichiro Ishige[3], Hiroki Toyoshima[2] & Ami Miyata[2]

Lipid astaxanthin, a potent antioxidant known as a natural sunscreen, accumulates in eukaryotic microalgae and confers photoprotection. We previously identified a photo-oxidative stress-inducible water-soluble astaxanthin-binding carotenoprotein (AstaP) in a eukaryotic microalga (*Coelastrella astaxanthina* Ki-4) isolated from an extreme environment. The distribution in eukaryotic microalgae remains unknown. Here we identified three novel AstaP orthologs in a eukaryotic microalga, *Scenedesmus* sp. Oki-4N. The purified proteins, named AstaP-orange2, AstaP-pink1, and AstaP-pink2, were identified as secreted fasciclin proteins with potent $^1O_2$ quenching activity in aqueous solution, which are characteristics shared with Ki-4 AstaP. Nonetheless, the absence of glycosylation in the AstaP-pinks, the presence of a glycosylphosphatidylinositol (GPI) anchor motif in AstaP-orange2, and highly acidic isoelectric points (pI = 3.6–4.7), differed significantly from that of AstaP-orange1 (pI = 10.5). These results provide unique examples on the use of water-soluble forms of astaxanthin in photosynthetic organisms as novel strategies for protecting single cells against severe photooxidative stresses.

---

[1] Department of Molecular Microbiology, Tokyo University of Agriculture, 1-1-1 Sakuragaoka, Setagaya-ku, Tokyo 156-8502, Japan. [2] Department of Bioscience, Tokyo University of Agriculture, 1-1-1 Sakuragaoka, Setagaya-ku, Tokyo 156-8502, Japan. [3] NODAI Genome Research Centre, Tokyo University of Agriculture, 1-1-1 Sakuragaoka, Setagaya-ku, Tokyo 156-8502, Japan. ✉email: kawashin@nodai.ac.jp

Photooxidative stress leads to the generation of reactive oxygen species (ROS) and results in critical damage in plants[1,2]. Plants use several systems to cope with photooxidative stress by scavenging ROS or protecting themselves from sunlight[3–5]. Carotenoids, which are known as natural lipid pigments, participate in the protective photooxidative stress mechanisms in plants by using their ability to deactivate triplet chlorophyll and singlet oxygen[6,7]. Carotenoids accumulate as lipid droplets also in microalgal cells, for example, astaxanthin in *Haematococcus*[8–10] and β-carotene in *Dunaliella*[11,12], assisting in ROS scavenging and acting as sunscreen.

Carotenoids can sometimes function in aqueous solution through chemical modifications or by binding proteins[13,14]. In photosynthetic organisms, two water-soluble carotenoid binding proteins, the orange carotenoid binding proteins (OCPs) and the peridinin-chlorophyll binding proteins (PCPs), have been identified and well-characterized in cyanobacteria and in dinoflagellates, respectively. OCPs are widely distributed in cyanobacteria and binds the 3′-hydroxyeqinenone, which plays role in the dissipation of light energy by interacting with phycobilisomes[15–17]. PCPs bind chlorophyll *a* and peridinine, are localized in the thylakoid lumen, and are distributed only in dinoflagellates. Their function is to transfer light energy to light-harvesting complexes[18,19].

Astaxanthin, a powerful antioxidant, is known to accumulate as lipid droplets in eukaryotic microalgae and protects cells under unfavourable environmental conditions[8–10,20–22]. Although water-soluble astaxanthin forms were unknown in photosynthetic organisms, a unique photooxidative stress-inducible water-soluble astaxanthin-binding protein, named AstaP, was identified in a eukaryotic microalga, *Coelastrella astaxanthina* Ki-4, isolated from a dry asphalt surface in midsummer in our previous study[23,24]. To our knowledge, this protein was the first one found in photosynthetic organisms that showed astaxanthin solubilization and accumulation. To our information, no other related proteins have yet been reported. In this study, we found novel AstaP orthologs produced by a microalga isolated from soil in a pond. Although these AstaP orthologs bound astaxanthin and were highly induced by photooxidative stress conditions, marked differences were found in their protein characteristics. The possible functions of AstaP proteins under photooxidative stress conditions and the distribution of related proteins in other organisms are discussed.

## Results

**An algal strain accumulates water-soluble pigments**. Okinawa-4N (Oki-4N), a eukaryotic microalgal strain isolated from a pond in 2002, was found to produce large amounts of aqueous pigments under photooxidative stress conditions, which was similar to that of *Coelastrella astaxanthina* Ki-4[23,24]. The 18S rRNA gene and internal transcribed spacer 2 (ITS2) gene sequences revealed that Oki-4N belongs to the family Scenedesmaceae of the order Chlorococcales and was placed in the *Scenedesmus* sensu stricto clade[24,25], which includes the type species of the genus *Scenedesmus*, *Scenedesmus obtusus* Meyen[26,27]. Based on genotypical and morphological characterization (Fig. 1a, b and Supplementary Figs. 1–3), we tentatively named this strain *Scenedesmus* sp. Oki-4N.

Under water-stress conditions (i.e., high salt or dehydration), Oki-4N cells were subjected to photooxidative stress by exposure to high light (HL, 800 μM photons m$^{-2}$ s$^{-1}$). When grown under these photooxidative stress conditions, the cells changed colour from green to brown and produced water-soluble pigments (Fig. 1a and Supplementary Fig. 3). The cells did not change colour under HL conditions without water stress or under low light (LL) conditions (50 μM photons m$^{-2}$ s$^{-1}$) with water stress. The rate of photosynthesis was reduced, but remained stable for the duration of the photooxidative stress treatments (Supplementary Fig. 4). In addition, *Chlamydomonas reinhardtii*, which is a model chlorophyte, showed chlorosis under these photooxidative stress conditions[23].

**Water-soluble carotenoid pigments of different colours**. Orangish pigments were detected by microscopic observation of cells, and the pigments were found to cover the cell sphere or to be localized in small vesicles (Fig. 1a and Supplementary Fig. 3). The stressed algal cells were disrupted in water-based buffer (50 mM Tris-HCl at pH 7.5), and the aqueous orange pigments were obtained by ultracentrifugation at $100,000 \times g$. The colour density of the aqueous orange pigments in the cell-free extracts (CFEs) from 4-day-stressed cells was spectroscopically measured and presented OD$_{480}$ value of about 0.7–1.0 cm$^{-1}$ (1 g of wet cells/10 mL cell suspension buffer, $n = 4$, Fig. 1c), and was estimated to correspond to OD$_{480}$ value of about 9–13 cm$^{-1}$ in the aqueous phase of the cell, taking into account the cellular water content. As the stress exposure period progressed, the colour density increased gradually, reaching the OD$_{480}$ value of about 20–35 cm$^{-1}$ (estimated concentration in the aqueous phase of the cell, $n = 4$) after two-weeks of stress treatment (Fig. 1a and Supplementary Fig. 3).

The aqueous orange pigments were evaluated by size exclusion column chromatography, and three pigments of different colours were visually fractionized (Fig. 1d). The first peak fraction (Peak-1) presented a yellowish colour, and its molecular size was estimated to be bigger than 670 kDa by gel filtration chromatography with top absorption peaks at 454 nm and 480 nm and a minor peak at 674 nm (Supplementary Fig. 5). The binding pigments were identified by HPLC and LC/MS analysis as chlorophyll *a* and *b*, β-carotene, and secondary carotenoids. Based on the presence of chlorophylls, we predicted that the Peak-1 pigment was derived from chloroplasts. The second peak fraction (Peak-2) showed orange colour and an estimated native molecular mass of around 75 kDa. A third peak fraction (Peak-3) showed a bright pink colour, and its estimated native molecular mass was around 20 kDa. The expressed amounts of Peak-2 and Peak-3 gradually increased as the duration of photooxidative stress progressed (Supplementary Table 1). These pigment peaks were also detected under dehydration conditions w/HL but were not detected under non-photooxidative stress conditions, including 0.25–0.5 M NaCl w/LL or dehydration w/LL. Under these experimental conditions, the superoxide dismutase (SOD) activity, which is one of the enzymes responsive to oxidative stress[3,28,29], increased under salt stress conditions w/HL but not w/LL (Supplementary Fig. 6).

**Orange and pink astaxanthin-binding proteins**. By using DEAE-sepharose column chromatography, one orange and two pink fractions were separated (Fig. 2a). Each pigment was further purified using a size exclusion column, and the purity and the molecular size of the purified proteins were determined by SDS-PAGE (Fig. 2b). The binding pigments (P1–P4) of each protein were found to be identical to those of AstaP-orange1[23] and were predicted to be astaxanthin, adonixanthin, lutein, and canthaxanthin (Fig. 2c, Supplementary Fig. 7). Based on the HPLC retention time, spectral features, molecular mass (determined with high-resolution LC/MS), and in silico analysis (using MS-MS spectra), P1 was determined to be astaxanthin. The binding ratio of astaxanthin to the total amount of binding carotenoids in each protein was estimated to be more than 85%. Therefore, we named these proteins AstaP-orange2, AstaP-pink1, and AstaP-pink2.

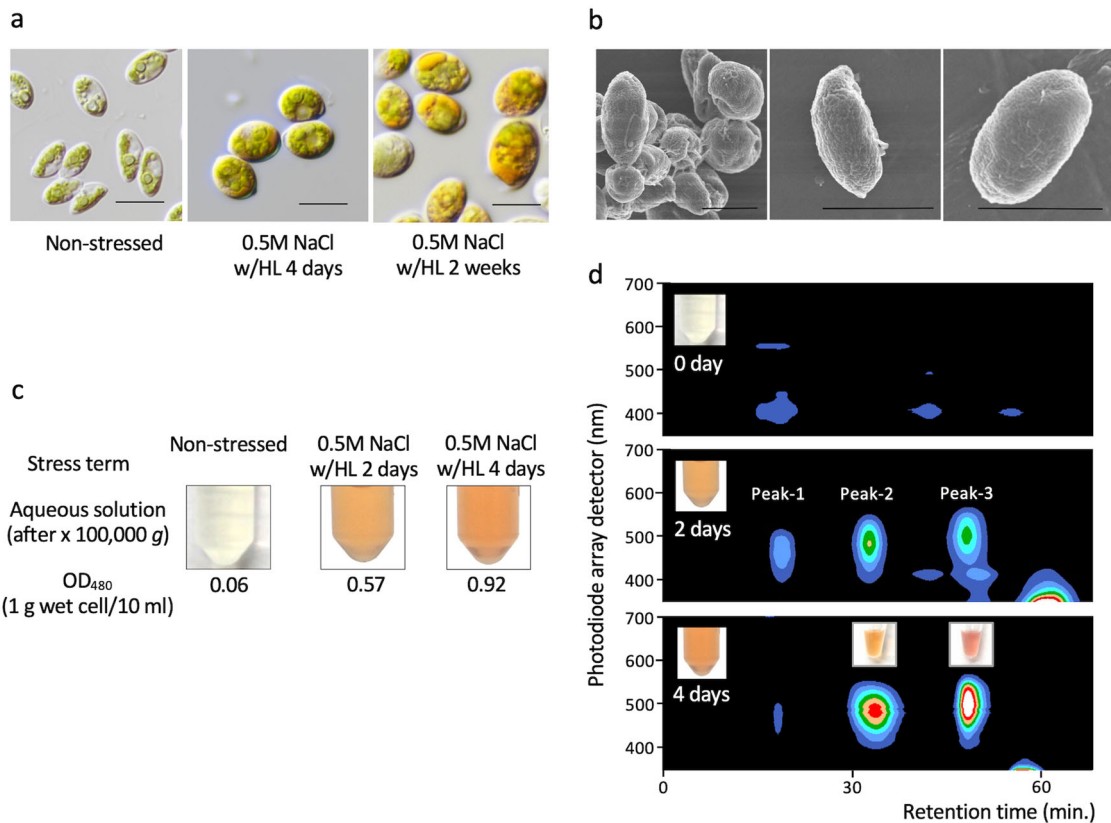

**Fig. 1 Identification of aqueous pigments in *Scenedesmus* sp. Oki-4N under photooxidative stress conditions. a** Light microscope view of Oki-4N. Cells were grown under non-stressed (left) or 0.5 M NaCl with high light (w/HL) for 4 days (middle) conditions and were continued for 2 weeks (right). Scale bar = 10 μm. **b** Scanning electron micrographs of Oki-4N. Scale bar = 10 μm. **c** Colour of aqueous cell extracts obtained after ultracentrifugation at 100,000 × *g* for 3 h. A 1 g aliquot of wet cells (the cell pellet after harvesting by centrifugation was lightly dried with filter paper to remove excess liquid medium) was suspended in 10 mL of cell suspension buffer (50 mM Tris-HCl at pH 7.5), disrupted by French press, and then ultra-centrifuged. Pigment concentrations were measured as optical density (OD) at 480 nm. The data are representative of at least three independent experiments. **d** Elution profiles of aqueous cell extracts from the cells under non-stressed or 0.5 M NaCl w/HL conditions for 2 or 4 days were evaluated by gel-filtration column chromatography. Elution profiles were monitored by using an HPLC photodiode array detector. The colour of the elution peak is shown above each elution peak. The data are representative of at least three independent experiments.

These AstaP orthologs were able to dissolve in ultra-pure water and showed the absorption maxima ($\lambda_{max}$) at 485 nm (AstaP-orange2), 502 nm (AstaP-pink1), and 501 nm (AstaP-pink2; Fig. 2d). These results indicated that the two pink AstaP proteins caused a bathochromic shift in orange carotenoids resulting in a pink colour and that the spectral shift was much shorter than the observed blue shifts in crustacyanins ($\lambda_{max}$ 630 nm)[14,30].

**Structure of AstaP-Orange2, AstaP-Pink1, and AstaP-Pink2.** The N-terminal amino acid sequence of the two purified pink proteins were determined (Fig. 3a). As N-terminal amino acid sequencing of purified AstaP-orange2 was not possible, peptide mass fingerprint (PMF) analysis was performed using a LC/MS/MS system. The cDNA sequences that encode each protein fragment were found in the cDNA sequence data from the photooxidative-stressed Oki-4N cDNA library. Each full-length cDNA clone was successively amplified by PCR, the nucleotide sequence was confirmed by genome sequencing (Supplementary Fig. 8), and the genes encoding AstaP-orange2, AstaP-pink1, and AstaP-pink2 were designated as *astaP-or2*, *astaP-pn1*, and *astaP-pn2*, respectively.

Each deduced amino acid sequence yielded N-terminal hydrophobic signal peptides for transmembrane secretion, and H1 and H2 domains conserved in the fasciclin protein family[31–34]

(Fig. 3a, b). The AstaP-orange2 had thirteen putative N-linked glycosylation sites (Asn-X-Thr), whereas AstaP-pink1 and AstaP-pink2 had no potential N-glycosylation sites. The presence of glycosylation in each protein was confirmed by PAS staining, used to detect polysaccharides (Fig. 2b). The isoelectric point (pI) estimated from the amino acid composition of AstaP-orange2, AstaP-pink1, and AstaP-pink2 were 3.6, 4.7, and 3.8 (calculated for mature proteins without signal sequences), respectively. AstaP-pink1 and AstaP-pink2 showed 89.5% sequence identity, and the sequence of AstaP-pink1 showed 41.4% and 41.9% identity with that of AstaP-orange1 and AstaP-orange2, respectively, in the overlapping regions (Supplementary Fig. 9). AstaP-orange2 contained two H1 and H2 domains, and each located tandemly within a protein (Fig. 3b). The front and rear parts of AstaP-orange2 showed 43.1% and 40.8% sequence identity with AstaP-orange1 in the overlapping region, respectively. (Supplementary Fig. 10).

These results indicated that the three AstaP proteins shared common properties, which include astaxanthin binding, acidic PI, fasciclin domains, and photooxidative stress-inducibility. However, several characteristics were different, namely, the presence of glycosylation sites, length of N-terminal signal sequences for transmembrane secretion, C-terminal region structure, bathochromic shifts to pink colour, and molecular size. The two AstaP-pink proteins were about 30 amino acids shorter at the

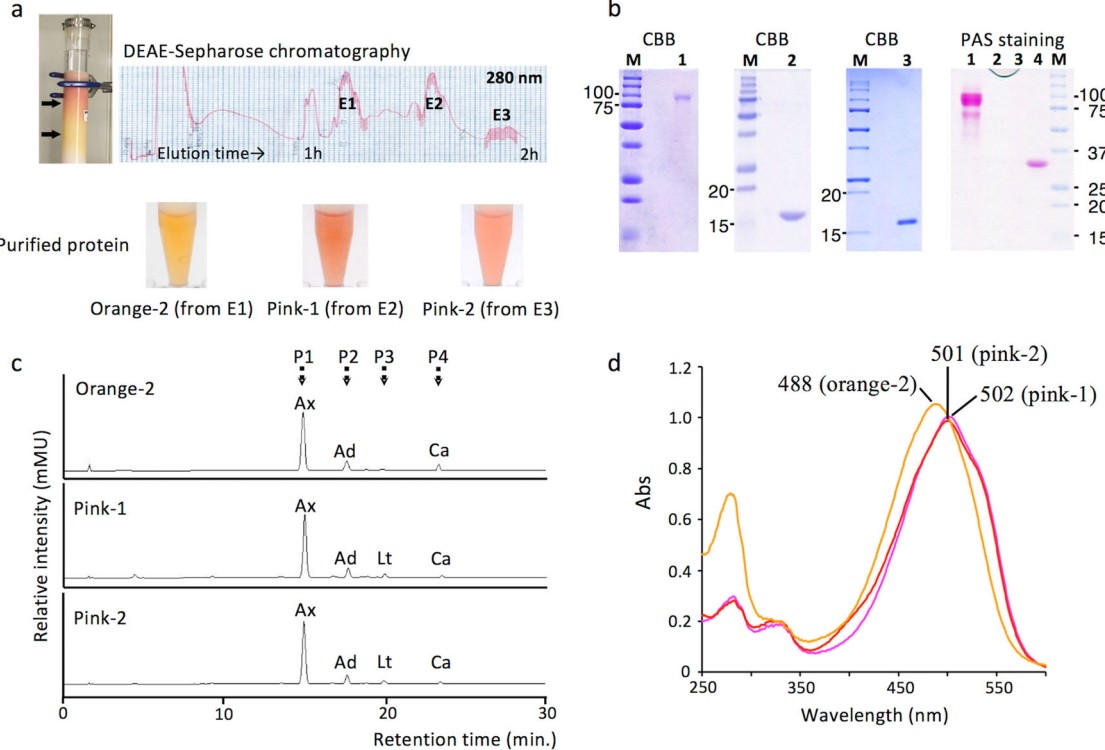

**Fig. 2 Purification of aqueous pigments. a** Elution profiles of aqueous pigments by DEAE sepharose column chromatography. Charged pigments were preserved in the upper layer of the column (left). The preserved pigments were eluted by an NaCl gradient. Arrows indicate the elution process for orange (lower arrow) and pink (upper arrow) pigments. The protein elution chart which was monitored at 280 nm is shown. The colour of the purified proteins from each eluted peak (E1–E3) are shown below the elution chart. **b** SDS-PAGE of the purified proteins stained with Coomassie Brilliant Blue. The purified proteins of orange-2 (lane 1), pink-1 (lane 2), and pink-2 (lane 3) are stained along with their corresponding molecular masses (kDa). The data are representative of at least three independent experiments. The picture on the right shows the PAS staining of the purified AstaPs. Lane 1, AstaP-orange2; lane 2, AstaP-pink1; lane 3, AstaP-pink2; lane 4, purified AstaP-orange1. M, pre-stained protein molecular mass standards (kDa). **c** HPLC elution profiles of the binding carotenoids from the purified AstaP-orange2, AstaP-pink1, and AstaP-pink2 proteins. P1–P4 indicate each elution peak. Ax: astaxanthin, Ad: adonixanthin, Lt: lutein, and Ca: canthaxanthin. The data are representative of at least three independent experiments. **d** Spectrum of the purified AstaP-orange2 (orange line), AstaP-pink1 (pale pink line), and AstaP-pink2 (red line). Each top peak wavelength is indicated. The data are representative of at least three independent experiments.

C-terminal region than Ki-4 AstaP-orange1 (Supplementary Fig. 9b). On the contrary, AstaP-orange2 has a C-terminal hydrophobic region composed of 30 amino acids, and it was absent in Ki-4 AstaP-orange1. The C-terminal region in AstaP-orange2 was predicted to be a glycosylphosphatidylinositol (GPI) lipid anchor motif for targeting the outer-plasma membrane, suggesting that it is a cell surface protein (Fig. 3a, b and Supplementary Fig. 10b)[35].

Phylogenetic analysis classified the AstaP proteins into two different clades, an AstaP-pink clade and an AstaP-orange clade, that showed sister relationships (Fig. 3c). AstaP orthologs were found in several eukaryotic microalgae, such as *Chlorella*, *Chlamydomonas*, and *Botryococcus* species, and all the proteins of unknown function had conserved fasciclin H1 and H2 domains. Since these microalgae are known to possess carotenoid synthesizing ability, it is considered that these AstaP orthologs possess similar functions.

**Singlet oxygen-scavenging activity in water-based solution.** The expression of AstaP-orange2 and both AstaP-pinks was considerably induced under photooxidative stress conditions (Fig. 1d, Supplementary Table 1). Northern blot analysis showed that the genes encoding AstaP-orange2 and the two AstaP-pinks were upregulated in the early phase of photooxidative stress and these inductions were enhanced after 4 days of stress treatment (Fig. 4a). AstaP proteins possess the ability to quench $^1O_2$ in

aqueous solution, and 0.3 μM of AstaP, which corresponds to the concentration of binding carotenoids, resulted in a quenching activity of 0.5–1.0 mM NaN₃, a well-known chemical $^1O_2$ quencher (Fig. 4b), which was similar to the activity of AstaP-orange1[23].

**Expression profiles of the AstaP proteins.** We found that the cells subjected to 0.25 M and 0.5 M NaCl stress w/HL for two weeks accumulated AstaP-orange2 and AstaP-pinks, respectively (Fig. 4c). Cells that were subjected to 0.25 M NaCl w/HL maintained photosynthesis, indicating that they were in a vegetative state (Fig. 4d and Supplementary Figs. 3, 4). Cells that were subjected to 0.5 M NaCl w/HL significantly reduced photosynthesis but retained cell components, such as pyrenoids and chloroplasts, and were considered to be in the cyst stage. Confocal microscopy analysis indicated that these pigments were not localized in chloroplasts (Fig. 4d). Although the purified AstaP proteins did not present detectable fluorescence under the experimental condition of 488 nm excitation/525–535 nm emission, small vesicles containing pigments from the 0.5 M NaCl treatment presented notable fluorescence. Based on microscopic and bioinformatic data (mainly on the presence of a C-terminal GPI-anchor motif with a hydrophobic region), the possible localization of AstaP-orange2 is likely to be the outer-plasma membrane. Contrarily, the lack of a GPI-anchor and protein

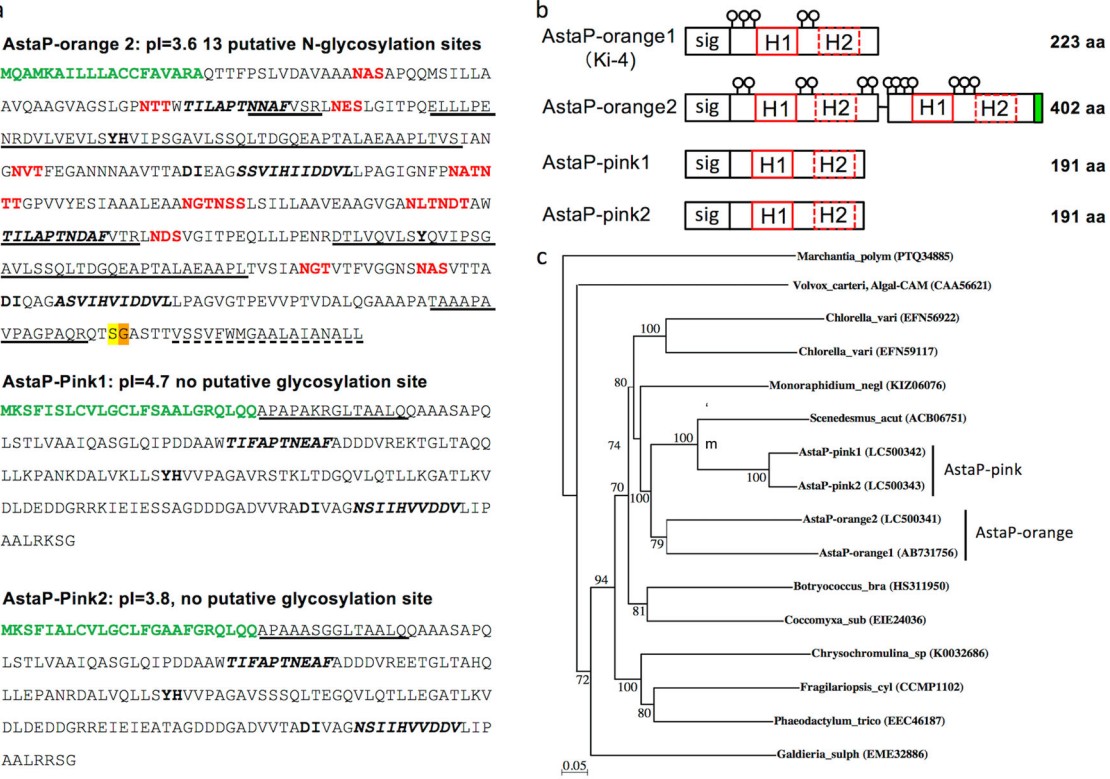

**Fig. 3 Purified AstaP proteins belong to the fasciclin protein family. a** The deduced amino acid sequence of the AstaP proteins. N-terminal signal peptides are shown in green fonts. Matched peptides by PMF analysis are underlined (AstaP-orange2). Experimentally detected N-terminal sequences are underlined (AstaP-pink1&2). H1 and H2 motifs of the fasciclin domains are shown in boldface italics. Potential sites for *N*-linked glycosylation are shown in red fonts. Y(H) and DI motifs, which are shown in boldface, are highly conserved in fasciclin protein family[31]. The predicted C-terminal hydrophobic GPI anchor signal sequence was dot underlined, and potential (yellow font) as well as alternative (orange font) GPI-modification sites are shown. **b** Comparisons between the AstaP protein structures. Red boxes represent highly conserved H1 and H2 fasciclin domains. The signal peptides (sig) are boxed. Circle pins indicate potential N-glycosylation sites. Green box represents a potential GPI anchor. **c** Neighbour-joining phylogenetic tree of the deduced sequence of the AstaP proteins with its homologues. A livewort homologue (Marchantia_polym: *Marchantia polymorpha*) was used as an outgroup, and Volvox Algal-CAM (Volvox_carteri: *Volvox carteri*) was used as a reference. Chlolella_vari: *Chlorella variabilis*, Monoraphidium_negl: *Monoraphidium neglectum*, Scenedesmus_acut: *Scenedesmus acutus*, Botryococcus_bra: *Botryococcus braunii*, Coccomyxa_sub: *Coccomyxa subellipsoidea*, Fragilariopsis_cyl: *Fragilariopsis cylindrus*, Phaeodactylum_trico: *Phaeodactylum tricornutum*, and Galdieria_sulph: *Galdieria sulphuraria*. The bootstrap values >50 are indicated at the branch points. Accession numbers for each protein are shown in parenthesis.

glycosylation in AstaP-pinks suggest that they are not localized to the cell surface, but to the small vesicles, such as vacuoles and endoplasmic reticulum, which was supported by the confocal microscopy data.

## Discussion

Microalgal astaxanthin is known to accumulate in extraplastidic lipid vesicles and is usually extracted as a lipophilic mono- or di-esterified form[36,37]. Although water-soluble form of astaxanthin were unknown in photosynthetic organisms, four types of water-soluble astaxanthin-binding proteins were found in Scene-desmaceae microalgae. These proteins were classified as members of the fasciclin family. Fasciclin domains are conserved in secretory and membrane-anchored proteins in humans, higher plants, and microalgae, such as fasciclin I in *Drosophila*[38], human periostins[39], plant arabinogalactan proteins[40] and SOS5[41], and the volvox cell adhesion molecule Algal-CAM[33]. These fasciclin family proteins have recently been found to be involved in multiple functions that are not limited to cell adhesion and proliferation, and they are also associated with multiple aspects of human health and disease[42], plant reproduction[43], and microbial stress responses, including a high-light response found in a cya-nobacterial ortholog[44]. To our knowledge, none of these proteins have been reported to bind lipids, including carotenoids, or to possess antioxidative activity. The lipid binding abilities of the fasciclin proteins and the distribution of AstaP orthologs in other organisms are currently being investigated in this study.

Although the benefit of using lipid astaxanthin in aqueous solution are still somewhat unclear, since astaxanthin has been shown to have a potent antioxidant activity, and its spectrum partially covers the area of spectrum peak of solar radiation at the sea level, we suggest that through AstaP, lipid astaxanthin could be transported to specific subcellular sites in the cell. Once there, it can provide sunshade effect and protect the light-sensitive biomolecules with its singlet oxygen-scavenging activity. The Oki-4N strain produces three types of AstaP proteins of unique properties. AstaP-pinks have been suggested to be localized in small vesicles, and they might function like the water-soluble anthocyanins in plants. AstaP-orange2, a glycosylated protein with a GPI-anchor motif, is expected to contribute in the trans-portation of astaxanthin to the cell surface. *Haematococcus* lipid astaxanthin was also shown to migrate to the cell surface under the strong sun light conditions[10]. This behaviour was reported to be astaxanthin specific and the movement has not been found in other pigments (lutein and β-carotene). These results suggest that compared to the other carotenoids, astaxanthin is the preferred xanthophyll used by these microalgae for photoprotection. It is

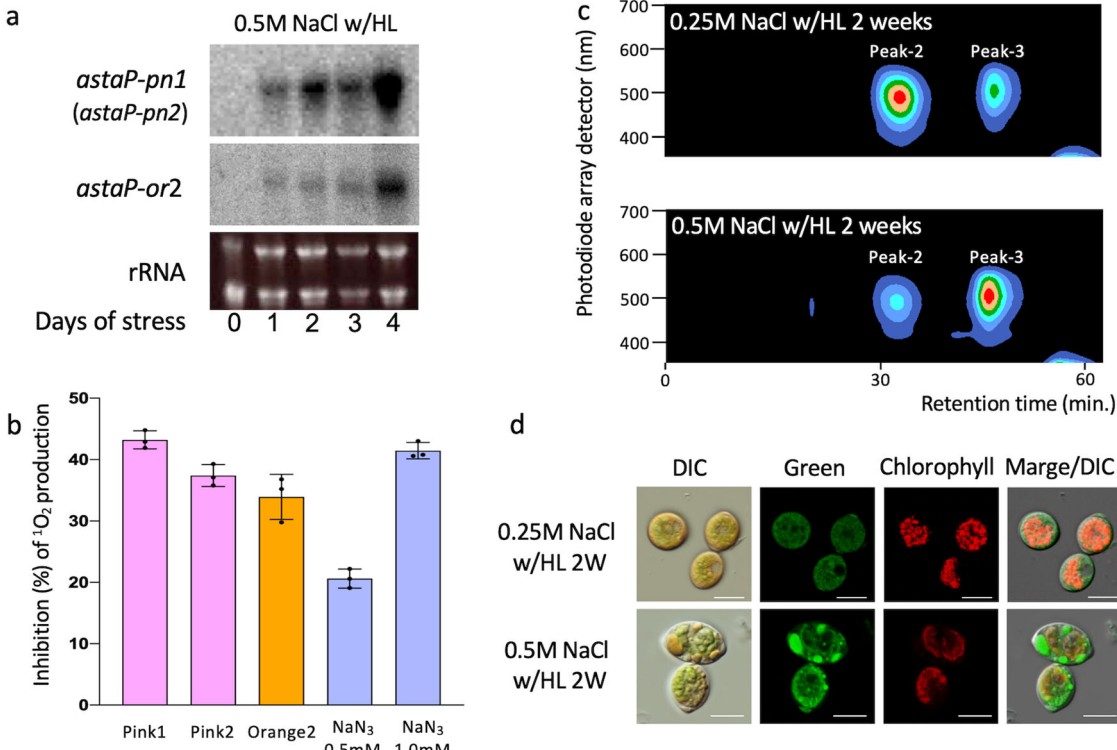

**Fig. 4 Expression profiles of the *astaP* genes in the early stage of photooxidative stress and characterization of the AstaP proteins. a** Northern blots of total RNA probed with the gene encoding AstaP-pink1 (*astaP-pn1*) and AstaP-orange2 (*astaP-or2*) amplified by PCR. The RNA encoding AstaP-pink2 (*astaP-pn2*) might be detected by the *astaP-pn1* probe due to its high sequence similarity. The Oki-4N strain was cultivated under high light conditions (w/HL) and subjected to 0.5 M NaCl with high light (w/HL) exposure. 0, just before the start of stress; 1–4, after 1, 2, 3, and 4 days of stress, respectively. Ethidium bromide staining of the ribosomal RNA (rRNA) to confirm equal RNA loading is shown below the autoradiogram. The membranes were reused for re-probing with different probes. The data are representative of two independent experiments. **b** Singlet oxygen quenching activity of the AstaP proteins. $^1O_2$ production was detected using a fluorescent probe SOSG. White light was irradiated for solutions containing (each final concentration) 5 μM SOSG, 1 μM rose bengal, and 0.3 μM of the AstaP proteins (final concentration of the binding carotenoids), or NaN₃ (0.5 mM or 1.0 mM) in a total volume of 500 μl. Data represent the mean of three independent measurements, and the error bars represent standard deviations. **c** Comparison of the expression of the AstaP proteins under different stress conditions. Oki-4N was cultivated under photooxidative stress conditions, with 0.25 M NaCl and high light exposure (w/HL), or 0.5 M NaCl and high light exposure for two weeks, respectively. Each aqueous cell extract obtained after ultracentrifugation at 100,000 × *g* for 3 h was loaded into a gel-filtration column for chromatography. The data are representative of three independent experiments. **d** Confocal microscopic analysis for estimating the cellular localization of the AstaP proteins in moderately stressed (0.25 M NaCl for 2 weeks) and highly stressed (0.5 M NaCl for 2 weeks) cells grown under high light exposure conditions. Green fluorescence (excitation 488 nm/emission 525–535 nm) under the same fluorescence intensity was pseudocoloured in green. Red fluorescence used to detect chlorophyll autofluorescence (excitation 633 nm/emission 670–705 nm) is pseudocoloured in red. Scale bar = 10 μm.

still unclear why astaxanthin is the number one choice for photoprotection, but we believe that AstaP's selective binding to astaxanthin holds a clue in elucidating the functional significance of astaxanthin as a specific protector against photooxidative stress.

Unlike AstaP proteins, other water-soluble carotenoproteins, including the astaxanthin-binding crustacyanins in lobster[45] or human retina and silkworm lutein-binding proteins[46,47], are generally fixed in shells or tissues. The 3′-hydroxyechinenon-binding cyanobacterial OCPs and dinoflagellate PCPs are associated with photosynthetic apparatus, and their functions and subcellular localization are different from AstaP. Based on the phylogenetic analysis, AstaP proteins were found to be a new member of the fasciclin protein family, which is composed of proteins with the ability to bind to astaxanthin with different protein structures. To understand the unique functions of AstaP proteins in organisms, a more detailed functional description, that should include the biological significance of their extraordinarily different pI, and the conformation of astaxanthin binding with fasciclin domains, is needed. In addition to their

physiological novelty, the biotechnological advantages of a mass producible and valuable astaxanthin in water-soluble form are likely to be of industrial interest.

## Methods

**Microalgal strains and growth conditions**. The Oki-4N strain was isolated from a pond soil sample in the Okinawa prefecture, Japan, and deposited in the Nodai Research Collection Centre (NRIC, a member of the World Federation for Culture Collection) as strain number NRIC 0987. The Oki-4N strain was permanently cryopreserved in NRIC. The composition of the A3 medium was the same as previously described[23]. The 18S rRNA gene and ITS2 sequence were analysed as previously described[23,24]. Briefly, the 18SF1 (5′-TAATGATCCTTCCGCAGGTT-3′) and 18SR1 (5′-CCTGGTTGATCCTGCCAG-3′) primers were used to amplify and sequence the 18S rDNA fragments. The primers ITS-F1 (5′-GGAAG TAAAAGTCGTAACAAGG-3′) and ITS-R1 (5′-TCCTCCGCTTATTGATATGC-3′) were used to amplify and sequence the ITS region. The amplified PCR products were sequenced at Macrogen (Macrogen Inc., Seoul, Korea) by capillary sequencing using a 3730xl DNA analyser (Applied Biosystems, Foster City, CA) on both strands. The multiple sequence data were initially aligned using MEGA ver. 7.0.2[48] and were subsequently manually refined. For 18S rDNA and ITS2 phylogenetic analyses, positions with deletions in most sequences were removed from the alignments, yielding 1,748 and 262 unambiguously aligned positions, respectively[23]. Phylogenetic trees were constructed using the neighbour-joining (NJ)

method using MEGA version 7.0.2. The reliability of the nodes for NJ analysis was estimated by bootstrapping with 1,000 replicates.

**Stress treatment**. Salt stress was induced by the addition of sterilized NaCl when growth reached an $OD_{750}$ value of 1.0 cm$^{-1}$ under conditions of high light exposure (~800 μmol photons m$^{-2}$ s$^{-1}$). Low light exposure (~50 μmol photons m$^{-2}$ s$^{-1}$) was used for comparison.

**Microscopy**. For light microscopy, we used a Nikon Eclipse E200 microscope (Nikon, Tokyo, Japan) equipped with differential interference contrast. For SEM analysis, the cells were initially fixed in 1% glutaraldehyde for 12 h, rinsed twice with phosphate-buffered saline at pH 7.4 (Thermo Fisher Scientific, Tokyo, Japan), and then fixed in 1% osmium tetroxide ($OsO_4$) at 25 °C. The samples were thereafter sequentially dehydrated in an ethanol gradient series at 50%, 70%, 80%, 90%, and 100% (v/v). The samples were dried by critical point drying, coated with osmium vapour using an osmium plasma coater, and observed using an S4800 scanning electron microscope (Hitachi Ltd., Tokyo, Japan) as previously described[23].

**Purification of AstaPs**. Salt stress was induced by the addition of 0.5 M NaCl when growth reached an $OD_{750}$ value of 1.0 cm$^{-1}$, and the cultures were continued for several days under high light conditions in a 5-L bottle. The harvested cells were suspended in 50 mM Tris-HCl buffer at pH 7.5, and CFEs were obtained by cell disruption in a French press (140 MPa) followed by the removal of the cell debris and lipids by centrifugation at $100,000 \times g$ for 3 h. Elution profile was monitored by using an HPLC photodiode array detector operated by LaChrome Elite software (Hitachi Ltd., Tokyo, Japan) for analysing elution profile and peak area. For purification, orange supernatants were passed through a DEAE-Sepharose Fast Flow column (GE Healthcare, Piscataway, NJ, USA). Fractions that contained orange and pink eluates were separately collected, and each fraction was passed through an HR100 gel filtration column (GE Healthcare). The orange or pink peak fractions were collected, and was subjected to SDS-PAGE to evaluate the purity. The molecular mass of the purified protein was determined by gel filtration chromatography and calibrated with the following molecular standards (Pharmacia, Uppsala, Sweden): ribonuclease (13.7 kDa), carbonic anhydrase (29 kDa), ovalbumin (44 kDa), conalbumin (75 kDa), aldolase (158 kDa), thyroglobulin (670 kDa), and blue dextran (2000 kDa) in three independent experiments. To test solubility in water, 500 μl of purified AstaP proteins were dissolved in 50 mM Tris-HCl buffer at pH 7.5 and dialyzed against 5 L of MilliQ (Millipore, Bedford, MA, USA) ultra-pure water for 6 h, and this procedure was repeated 3 times. Water solubility was also tested by passage through a PD-10 desalting column (GE Healthcare) eluted with MilliQ water. Periodic acid-Schiff (PAS) staining was performed with a commercially available staining kit (Merck, Darmstadt, Germany) according to the instructions from the manufacturer to determine protein glycosylation in PAGE gel.

**Isolation of RNA and construction of a cDNA library**. To prepare a cDNA library, Oki-4N cells that were subjected to salt stress for 1–6 days under high light conditions were harvested, and total RNA was extracted with Trizol reagent (Roche, Basel, Switzerland)[23]. Poly(A)$^+$ mRNA was isolated and used to generate a full-length cDNA library with a Smart-Infusion PCR cloning system (Clontech, Palo Alto, CA)[23]. All steps involved in cDNA synthesis were performed according to the instructions from the manufacturer. The cDNA libraries were sequenced using the Illumina HiSeq 2500 system (Illumina, San Diego, CA). The FASTQ files were imported to the CLC Genomics Workbench (Qiagen, Hilden, Germany), and de novo sequence assembly was performed. A total of $45 \times 10^3$ contigs were generated after the de novo assembly by the CLC software.

**Bioinformatics**. Bioinformatic analyses were performed as previously described[23]. Briefly, database searches for sequence homology were performed using the programs BLASTp (http://www.ncbi.nlm.nih.gov/BLAST/) and FASTA (http://www.genome.jp/tools/fasta/) set to the standard parameters. The N-terminal signal sequence was predicted using SignalP (http://www.cbs.dtu.dk/services/SignalP/). The *N*-linked glycosylation site was predicted by the program NetNGlyc 1.0 (http://www.cbs.dtu.dk/services/NetNGlyc/). The C-terminal hydrophobic GPI anchor signal sequence was predicted by big-PI Plant Predictor (http://mendel.imp.ac.at/gpi/plant_server.html). Protein sequences were aligned using ClustalW (http://clustalw.ddbj.nig.ac.jp/index.php?lang=en). The isoelectric point (pI) and molecular mass were calculated by GENETYX-MAC software (Genetyx Corporation, Tokyo, Japan). Protein sequences homologous to AstaP proteins were retrieved from NCBI BLASTp search, and top10 sequences from different species of eukaryotic microalgae were selected for making phylogenetic tree. A phylogenetic tree was constructed using the neighbour-joining (NJ) method using MEGA version 7.0.2. The reliability of the nodes for NJ analysis was estimated by bootstrapping with 1,000 replicates.

**Determination of peptide sequences**. The N-terminal amino acid sequence was determined by the Edman degradation method as previously described[23]. PMF

analysis was performed using LC/MS/MS and Peaks Studio 8.5 (Bioinformatics solutions Inc., Waterloo, Canada). Briefly, a piece of purified protein SDS-PAGE gel was excised, digested by trypsin (Promega, Madison, WI), and purified with a ZipTip C18 column (Millipore, Bedford, MA) according to the instructions from the manufacturer. Peptide masses were determined in the positive-ion reflector mode in an Orbitrap Q Exactive focus LC/MS/MS system (Thermo Scientific, Bremen, Germany) using a CAPCEL PAK C18 reversed-phase column (150 × 2.0 mm i.d., 5 μm particle size, Shiseido, Tokyo, Japan). PMF analyses were conducted with Peaks Studio 8.5. To confirm the nucleotide sequence of cDNAs encoding the N-terminal sequences of the AstaP-pinks and the trypsin-digested AstaP-orange2 peptides, we amplified each cDNA by PCR using the Oki-4N cDNA library as a template. The amplified PCR products were sequenced by juxtaposing sequencing from the 5′ and 3′ portions to confirm the full-length cDNA sequence. The genomic regions for each mRNA were sequenced to confirm the nucleotide sequence and genome structure.

**Pigment determination**. Pigments were analysed as previously described[23]. Briefly, the AstaP binding pigments (P1–P4 peaks) were determined based on the absorption spectra obtained using an HPLC photodiode array detector, HPLC retention times, and molecular masses from high-resolution LC/MS/MS analysis in comparison with those of standard compounds. Synthetic astaxanthin {(3S,3′S)-3,3′-Dihydroxy-β,β-carotene-4,4′-dione}, lutein {(3R,3′R,6′R)-β,ε-Carotene-3,3′-diol}, canthaxanthin (β,β-Carotene-4,4′-dione), β-carotene (β,β-Carotene), adonirubin (β,β-Carotene-3-hydroxy-β,β-carotene-4,4′-dione), and adonixanthin {(3S,3′S)-3,3′-dihydroxy-β,β-caroten-4-one} were obtained from Carotenature (Lupsingen, Switzerland). Astaxanthin, lutein, canthaxanthin, and β-carotene were also obtained from Wako Pure Chemicals (Osaka, Japan), and their HPLC profiles were confirmed to be consistent with those from Carotenature. An Orbitrap LC/MS/MS system (Thermo Scientific, Bremen, Germany) was used for high-resolution LC/MS/MS analysis. Positive and negative ion mass spectra of the column eluate were recorded in full scan mode (m/z 100–1500) with an electrospray ionization (ESI) source. The interface voltage was set to 4.5 or −3.5 kV. Nitrogen gas was used as the nebulizing gas at a flow rate of 1.5 L min$^{-1}$. Charged droplet and heat block temperatures were both 300 °C. Solvents were of LC/MS quality. The data were analysed by Compound Discoverer v 2.1 (Thermo Fisher Scientific, Bremen, Germany). The MS data (from representative data) of the P1 peak from Orange-2, Pink-1, and Pink-2 appeared to indicate [MH]$^+$ at m/z 597.3932, 597.3917, and 597.3941, ($C_{40}H_{52}O_4$, error <5 ppm), respectively. The MS-MS spectra of each P1 pigment was matched to that of astaxanthin in the mzCloud database using Compound Discoverer v. 2.1.

**Enzyme assay**. Measurement of $^1O_2$ production was detected using a fluorescent probe Singlet Oxygen Sensor Green (SOSG, Molecular Probes, Eugene, OR) as previously described[23]. Briefly, SOSG was dissolved in methanol to make a stock solution of 5 mM and then was diluted by MilliQ ultra-pure water to 0.5 mM. Fluorescence measurements were made with an RF6000 spectrofluorometer (Shimadzu, Kyoto, Japan) using an excitation/emission of 488/525 nm for a solution containing 5 μM SOSG, 1 μM rose bengal, and several concentrations of AstaP or NaN$_3$ in air-saturated 100 mM Tris-HCl buffer at pH 7.5 and 25 °C in a total volume of 500 μl. The concentration of the binding carotenoids in AstaP was estimated from the peak absorbance of each AstaP protein by using the absorption coefficient of astaxanthin ($\varepsilon_{476} = 125$ mM$^{-1}$ cm$^{-1}$)[49]. Samples were exposed to an EFD21ED (21W, 400–650 nm) white light (Toshiba, Tokyo, Japan) with a light intensity of 130 μmol photons m$^{-2}$ s$^{-1}$, resulting in the rose bengal-photosensitized generation of $^1O_2$. The inhibition of $^1O_2$ production by AstaP proteins or NaN$_3$ was measured. Data represent the mean of three independent measurements, and the error bars represent the standard deviations. Superoxide dismutase (SOD) activity was followed by monitoring the inhibition of cytochrome c reduction by superoxide generated by xanthine/xanthine oxidase at 25 °C as described[50,51]. One unit of SOD activity is defined as the amount of protein that inhibits the rate of cytochrome c reduction by 50%.

**Northern hybridization**. Total RNAs from cell lysates harvested at several time points after the induction of stress under high light conditions were isolated using TRIzol® (Invitrogen, Carlsbad, CA). Northern blotting was carried out by previously described standard procedures[23]. The RNA (10 μg) was subjected to electrophoresis in 1.0% agarose gel, blotted onto nylon Hybond N$^+$ membranes (Amersham, Piscataway, NJ), and the membranes were probed with the PCR-amplified DNA fragment encoding the target region. The identity of the amplified DNA fragment was confirmed by size and nucleotide sequencing. Following the pre-hybridization of the RNA at 60 °C for 30 min, the $^{32}$P-labelled DNA probe was hybridized to RNA on the membrane at 60 °C for 12 h. The membrane was then probed with the $^{32}$P-labeled gene probe of the full length *astaP-pn1* obtained by PCR. The membrane was subsequently stripped of the probe and re-probed with the full-length *astaP-or2* probe.

**Optical and confocal microscopy**. Oki-4N cells were observed on a Nikon Confocal Fluorescent Microscope (Tokyo, Japan) equipped with differential interference contrast and a ×60 water immersion objective. Green fluorescence was

detected in the range of 525–535 nm, and the excitation wavelength was set to 488 nm. AstaP did not show detectable green fluorescence under the experimental condition. Chlorophyll autofluorescence was detected in the range of 670–705 nm, and the excitation wavelength was set to 633 nm[52]. Images were acquired by using the Nikon FluoView FV3000 software (Tokyo, Japan).

**Statics and reproducibility**. Experiments were repeated multiple times as indicated in the figure legends to confirm the reproducibility of the data. Graphs were produced and statistical analyses performed using Prism version 8.4.3 (GraphPad Software, San Diego, CA, USA) and Microsoft Excel version 16.39.

**Reporting summary**. Further information on research design is available in the Nature Research Reporting Summary linked to this article.

## Data availability

The algal strains are available from the NRIC culture collection centre or the corresponding author upon reasonable request. The cDNA sequence and genomic sequence data of Asta-orange2, AstaP-pink1 and AstaP-pink2, and the 18S rRNA gene sequence and the ITS gene sequence of strain Oki-4N has been deposited in the DDBJ/EMBL/Genbank database under the accession numbers LC500341-LC500343, and LC500286-LC500287, respectively. All data used to support the findings of this study have been included in the paper and Supplementary information or made available through the National Centre for Biotechnology Information.

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

## Acknowledgements

The authors thank many colleagues for their advice, discussions, and technical assistance, especially Kenta Nakahodo, Kaoru Saji, Sayaka Nakajima, Kohei Omori, Dr. Takumi Satoh, and Prof. Youichi Niimura. The authors thank Prof. Shinichi Takaichi for useful advice on carotenoid analyses, and also thank Prof. Yukio Yaguchi and Yumi Goto for performing microscopic analyses. The authors thank Eri Kubota for performing RNA sequencing. This study was supported in part by a grant-in-aid from the Japan Society for Promotion of Science (No. 26440155, to S.K.), the Institute for Fermentation, Osaka (IFO, No. G-2014-2-072, to S.K.), and the Cosmetology Research Foundation (J-15-2, to S.K.), and MEXT-Supported Program for the Strategic Research Foundation at Private Universities (S1311017).

## Author contributions

S.K. conceived and designed this study. S.K., K.Y., T.N., H.K., and A.M. performed the experiments, analysis, and wrote the paper. T.I. performed RNA sequencing. All authors discussed the results and commented on the paper.

## Competing interests

The authors declare no competing interests.
