## [Peer Review File · Communications Biology]

Reviewers' comments:

Reviewer #1 (Remarks to the Author):

Review Report

Title: Photooxidative stress-inducible microalgal orange and pink water-soluble astaxanthin-binding proteins" by Prof Kawasaki and colleagues

Paper # COMMSBIO-20-0361-T

This paper reports the study of isolating astaxanthin-binding protein and examining its molecular biological, physiological and phylogenetic characteristics. Overall, it is well-written and acceptable, and the topic covered here will interest a wider audience. However, there are some questions and issues in this manuscript, and the authors should revise based on the comments below.

Comments:

L.29: "GPI" is the first abbreviation in the text, so please spell it out.

L.59: Please describe a little more about the "extreme" environment.

L.64-65: "significant difference" is generally used in the context of statistical tests. The authors should be described differently.

L.76: It is better to italicize "sensu stricto", but please follow the submission guidelines for details.

L.122: Spell out "DEAE".

L.125-L.128: When interpreting this sentence in two parts: "The binding pigments (P1-P4) of each protein were found to be identical to the previously described AstaP-orange1" and "(The binding pigments...) were predicted to be astaxanthin, adonixanthin, lutein, and canthaxanthin.: The latter text is true, but the former can be interpreted as "pigments = AstaP-orange1", which is a bit strange.

L.154: A brief description of "PAS" would be helpful to the readers.

L.190: Do the authors use "NaN3" as a standard quencher? A brief explanation of the role of NaN3 would make it easier to understand the data interpretation.

L.198-L.199: In the sentence "Confocal microscopy ... in chloroplasts (Fig. 4d)", it would be better to describe in more detail that AstaP has green autofluorescence. The same applies to L.413.

L.200: "significant differences" are generally used in the context of statistical tests, so the authors should be described differently.

L.205-L.207: The authors should quote the data corresponding to this description.

L.225: "in our laboratory" is better than "in this study" (but it is up to the authors to decide whether to change).

L.228: Please quote the citation for the description of "similar spectrum to that of the solar spectrum at sea level", if any.

L. 228-L.229: "we conclude ...in the cell." Is the discussion here intended for hydrophilic astaxanthin?

If so, it feels a bit strong to use "conclude" as there is no direct evidence of the subcellular transport to guide this conclusion. I recommend changing to ", suggesting that" or something that instead of "we conclude that".

L.231: Does "at the specific site" mean "subcellular specific site"?

L. 238: "b-" Replace "b" in caroten with Greek letters.

L.244-L254: In this study, the authors performed a molecular phylogenetic analysis of the AstaP proteins, so it is recommended that the authors add some insights from that analysis. How about inserting a brief discussion/speculation, for example, between "AstaP" and "To understand" in L.249?

L.353: Spell out the "PMF".

Reviewer #2 (Remarks to the Author):

The paper entitled "Photooxidative stress-inducible microalgal orange and pink water-soluble astaxanthin-binding proteins" by Kawasaki and coauthors identified three photooxidative stress-inducible water-soluble astaxanthin-binding carotenoprotein (AstaP) in microalga *Scenedesmus* sp. Oki-4N, named: AstaP-orange2, AstaP-pink1, and AstaP-pink2. The manuscript is well-prepared and data rich. However, I do not see the novelty of this study. AstaP is already described in detail in *Coelastrella astaxanthina* Ki-4 in 2013 by the leading author. The current study does not offer any new insight into the function of AstaP proteins. Similar to Ki-4 AstaP, the Oki-4N AstaPs are cytosolic proteins and are induced by oxidative plus salt stress. The purified AstaP proteins can quench singlet oxygen in vitro, demonstrating their possible in vivo function. Salt stress alone is enough for induction of AstaPs, however; oxidative stress (high light) alone does not lead to the induction of AstaP proteins. Given the important role of astaxanthin as antioxidant this result is puzzling. They measured rate of photosynthesis as proxy for oxidative status in high/low light, plus/minus salt stress and they see photoinhibition only in the presence of salt stress. Can the authors clarify the level of reactive oxygen species for the low/high light with/out salt stressed cells to identify the exact signal that leads to induction of AstaP proteins.

Reviewer #3 (Remarks to the Author):

The manuscript Kawasaki et al. reported the detailed analysis of water-soluble astaxanthin binding proteins (AstaP) in an alga *Scenedesmus* sp. Oki-4N. The authors purified three AstaP orthologs, named AstaP-orange2, -pink1, and -pink2 from the alga grown under stress conditions (high-salt and high-light). The detailed characterization of those three pigment proteins elucidated that each AstaP bound astaxanthin, and had their specific absorption feature. Further, the authors evaluated the singlet oxygen quenching activity of the AstaP proteins and found that all these AstaP proteins were capable to quench the single oxygen. Taken together with the gene expression and protein accumulation of AstaP only under stress conditions, the author envisioned that the astaxanthin-binding AstaP proteins might act as specific protectors against photooxidative stress. Overall, the biochemical and phylogenetic results of the AstaP proteins described in the manuscript sounds confident. The conclusion (hypothesis) made by the authors is plausible.

I have several questions about the results presented in the current manuscript.

1. Why the authors used 2 weeks stressed cells for Figure 4c/d, while 1-4 days stressed cells were used for Figure 1c/d/e and Figure 4a? From Figure 1c/d, the alga accumulated clear amounts of AstaP proteins within 2-4 days. Is there any reason to use 2 weeks stressed cells for Figure 4c/d experiments?
2. I found that there is only little correlation between gene expression and protein expression of AstaP proteins. Although AstaP proteins were clearly expressed within 2 days of stress treatment (Figure 1c/d/e), its genes seemed to be not much expressed at that time (Figure 4a). The AstaP genes seemed to be highly expressed after 4 days of stress treatment. I would like to suggest the authors perform qRT-PCR of the genes to show a more clear correlation between gene and protein expression of AstaP.
3. What is the control of the singlet oxygen quenching activity (Figure 4b)? I could not find this point in the Methods section of the manuscript. Because the AstaP proteins used in this experiment seemed to be resuspended in the purification buffer, the authors have to show the effect of the purification buffer on the singlet oxygen quenching activity. According to this, the authors should show control values.
4. What is the unit for "Relative photosynthesis" in Supplementary Figure 4? From the figure legend, I predicted that the authors measured the oxygen evolution activity of the cells. If so, please provide the original unit of the measurement (e. g. XXX mol O₂/cell). Or it might be better to adjust the initial values at "1" because I could not understand the reason why the authors adjusted the initial values at "about 2.4".

Our responses to reviewers' comments are as follows:

Responses to Reviewer #1

Reviewers' comments:

Reviewer #1 (Remarks to the Author):

Review Report

Title: Photooxidative stress-inducible microalgal orange and pink water-soluble astaxanthin-binding proteins" by Prof Kawasaki and colleagues
Paper # COMMSBIO-20-0361-T

This paper reports the study of isolating astaxanthin-binding protein and examining its molecular biological, physiological and phylogenetic characteristics. Overall, it is well-written and acceptable, and the topic covered here will interest a wider audience. However, there are some questions and issues in this manuscript, and the authors should revise based on the comments below.

Comments:

We thank you for the valuable comments. We have revised the manuscripts accordingly.

L.29: "GPI" is the first abbreviation in the text, so please spell it out.

Answer: We have defined the abbreviation in the text (line30).

L.59: Please describe a little more about the "extreme" environment.

Answer: We have provided more information accordingly (line 61).

L.64-65: "significant difference" is generally used in the context of statistical tests. The authors should be described differently.

Answer: We thank you for your suggestion. We have made the necessary change (line 66).

L.76: It is better to italicize “sensu stricto”, but please follow the submission guidelines for details.

Answer: We referred to several taxonomic papers and confirmed that ‘sensu stricto’ is not italicised.

L.122: Spell out “DEAE”.

Answer: DEAE-sepharose is the product name, and therefore, it is not necessary to spell it out.

L.125-L.128: When interpreting this sentence in two parts: “The binding pigments (P1-P4) of each protein were found to be identical to the previously described AstaP-orange1” and “(The binding pigments...) were predicted to be astaxanthin, adonixanthin, lutein, and canthaxanthin.: The latter text is true, but the former can be interpreted as “pigments = AstaP-orange1”, which is a bit strange.

Answer: We thank you for your comment. We have revised the sentence accordingly (line 131).

L.154: A brief description of “PAS” would be helpful to the readers.

Answer: We have added a brief explanation in the text (line 159).

L.190: Do the authors use “NaN₃” as a standard quencher? A brief explanation of the role of NaN₃ would make it easier to understand the data interpretation.

Answer: We have added a brief explanation in the text (line196)

L.198-L.199: In the sentence “Confocal microscopy ... in chloroplasts (Fig. 4d)”, it would be better to describe in more detail that AstaP has green autofluorescence. The same applies to L.413.

Answer: AstaP does not show a detectable green autofluorescence. We have added a brief explanation in the text (line 427). We replaced the images in Fig. 4d (for 0.25 M) to show the different types of cells in supplementary Figure3.

L.200: “significant differences” are generally used in the context of

statistical tests, so the authors should be described differently.

Answer: We have revised the word in the text (line 207).

L.205-L.207: The authors should quote the data corresponding to this description.

Answer: We have revised the sentence in the text (line 214-215).

L.225: “in our laboratory” is better than “in this study” (but it is up to the authors to decide whether to change).

Answer: We have revised the relevant sentence (line 233).

L.228: Please quote the citation for the description of “similar spectrum to that of the solar spectrum at sea level”, if any.

Answer: I am assuming that the sentence you pointed out did not represent accurate information. It is well known that the spectral peak of solar radiation is around 500 nm. According to this information, we have revised the relevant sentence (line 236).

L. 228-L.229: “we conclude ...in the cell.” Is the discussion here intended for hydrophilic astaxanthin? If so, it feels a bit strong to use “conclude” as there is no direct evidence of the subcellular transport to guide this conclusion. I recommend changing to “, suggesting that” or something that instead of “we conclude that”.

Answer: We have replaced ‘conclude’ with ‘suggest’ (line 237).

L.231: Does “at the specific site” mean “subcellular specific site”?

Answer: Answer: We have revised the relevant sentence (line 238).

L. 238: “b-“ Replace “b” in caroten with Greek letters.

Answer: We have made the necessary change (line 246).

L.244-L254: In this study, the authors performed a molecular phylogenetic analysis of the AstaP proteins, so it is recommended that the authors add some insights from that analysis. How about inserting a brief discussion/speculation, for example, between “AstaP” and “To understand” in L.249?

Answer: We have added a sentence based on the phylogenetic analysis (lines 257–259).

L.353: Spell out the “PMF”.

Answer: PMF has been defined in line 147.

Reviewer #2 (Remarks to the Author):

The paper entitled “Photooxidative stress-inducible microalgal orange and pink water-soluble astaxanthin-binding proteins” by Kawasaki and coauthors identified three photooxidative stress-inducible water-soluble astaxanthin-binding carotenoprotein (AstaP) in microalga *Scenedesmus* sp. Oki-4N, named: AstaP-orange2, AstaP-pink1, and AstaP-pink2. The manuscript is well-prepared and data rich. However, I do not see the novelty of this study. AstaP is already described in detail in *Coelastrella astaxanthina* Ki-4 in 2013 by the leading author. The current study does not offer any new insight into the function of AstaP proteins. Similar to Ki-4 AstaP, the Oki-4N AstaPs are cytosolic proteins and are induced by oxidative plus salt stress. The purified AstaP proteins can quench singlet oxygen in vitro, demonstrating their possible in vivo function. Salt stress alone is enough for induction of AstaPs, however; oxidative stress (high light) alone does not lead to the induction of AstaP proteins. Given the important role of astaxanthin as antioxidant this result is puzzling. They measured rate of photosynthesis as proxy for oxidative status in high/low light, plus/minus salt stress and they see photoinhibition only in the presence of salt stress. Can the authors clarify the level of reactive oxygen species for the low/high light with/out salt stressed cells to identify the exact signal that leads to induction of AstaP proteins.

Answer: We thank you for the valuable comment. Currently, microalgal astaxanthin is garnering attention as a functional pigment in various industries. They exist in microalgal cells in a lipophilic form; Ki4 AstaP is a water-soluble form, which was identified for the first time in photosynthetic organisms. As you pointed out, the AstaP proteins identified in this study showed similar characteristics as they bind to astaxanthin like Ki-4 AstaP. However, the structural features showed variations, including completely different pI, and the presence of glycosylation and GPI motif, and so, our study not only suggests the possibility of distribution of AstaP proteins in a wide

range of microalgal species but also provides novel information about their diverse intracellular functionalities.

As prokaryotic Cyanobacteria members perceive high light intensity-induced oxidative stress, even at the light intensity of 300 μ E, considerable physiological responses including the expression of OCP are observed. On the contrary, as eukaryotic Ki-4 and Oki-4N were isolated from environmental conditions with a high light, they have the ability to develop colonies even under the sunlight in midsummer. The expression of AstaP proteins was not observed under the condition of high light alone. In the present study, we measured the SOD activity, which is one of the indicators of photooxidative stress, and presented the results in the supplementary data. The results indicated that Oki-4N seems to perceive photooxidative stress under salt stress conditions with a high light intensity.

Reviewer #3 (Remarks to the Author):

The manuscript Kawasaki et al. reported the detailed analysis of water-soluble astaxanthin binding proteins (AstaP) in an alga *Scenedesmus* sp. Oki-4N. The authors purified three AstaP orthologs, named AstaP-orange2, -pink1, and -pink2 from the alga grown under stress conditions (high-salt and high-light). The detailed characterization of those three pigment proteins elucidated that each AstaP bound astaxanthin, and had their specific absorption feature. Further, the authors evaluated the singlet oxygen quenching activity of the AstaP proteins and found that all these AstaP proteins were capable to quench the single oxygen. Taken together with the gene expression and protein accumulation of AstaP only under stress conditions, the author envisioned that the astaxanthin-binding AstaP proteins might act as specific protectors against photooxidative stress.

Overall, the biochemical and phylogenetic results of the AstaP proteins described in the manuscript sounds confident. The conclusion (hypothesis) made by the authors is plausible.

I have several questions about the results presented in the current manuscript.

We thank you for your valuable comments. We have revised the manuscript accordingly.

1. Why the authors used 2 weeks stressed cells for Figure 4c/d, while 1-4 days stressed cells were used for Figure 1c/d/e and Figure 4a? From Figure 1c/d, the alga accumulated clear amounts of AstaP proteins within 2-4 days. Is there any reason to use 2 weeks stressed cells for Figure 4c/d experiments?

Answer: We provided the results from short-term-stressed cells in Fig. 1 because we wanted to show that the AstaP proteins have the ability to respond to the early stage of photooxidative stress. On the contrary, changes in the expression pattern of AstaP-pinks and AstaP-orange2 were observed under long-term stress conditions; the results suggested differences in the functionality of these proteins. Therefore, we presented the data obtained under 2-week stress conditions in Fig. 4. We have revised the relevant sentence to address your concern (line 196).

2. I found that there is only little correlation between gene expression and protein expression of AstaP proteins. Although AstaP proteins were clearly expressed within 2 days of stress treatment (Figure 1c/d/e), its genes seemed to be not much expressed at that time (Figure 4a). The AstaP genes seemed to be highly expressed after 4 days of stress treatment. I would like to suggest the authors perform qRT-PCR of the genes to show a more clear correlation between gene and protein expression of AstaP.

Answer: In this study, we performed northern blotting to observe the expression of the target genes. We have replaced the image in Fig. 4a with that obtained at a higher exposure time. We hope that you can clearly observe the gene expression, especially the difference between undetectable gene expression under non-stressed condition and induced gene expression under 1-day stress condition. Although qRT-PCR is one of the valuable tools to quantify gene expression, we did not prefer this method because it depends on amplification efficiency of the target gene, and the data may contain unspecific gene amplification from the background genes. Therefore, we used northern blotting to detect the specific expression of the target gene. As you commented, the data of high induction under 4-day stress condition is very interesting. We are thinking that the *astaP* gene might be responsive to several signals including photooxidative stress. It is well known that the changes in gene expression are frequently not reflected at the protein translation level, and this inconsistency depends on various biochemical regulation mechanisms. [REDACTED]

3. What is the control of the singlet oxygen quenching activity (Figure 4b)? I could not find this point in the Methods section of the manuscript. Because the AstaP proteins used in this experiment seemed to be resuspended in the purification buffer, the authors have to show the effect of the purification buffer on the singlet oxygen quenching activity. According to this, the authors should show control values.

Answer: We thank you for your valuable comment. We measured the SOSG fluorescence without adding AstaP or NaN_3 (buffer only) into the reaction cuvette, and used it as a control value; then, we measured the inhibition rate of SOSG fluorescence from the control value after adding quenchers. We have revised the relevant text in the methods section.

4. What is the unit for “Relative photosynthesis” in Supplementary Figure 4? From the figure legend, I predicted that the authors measured the oxygen evolution activity of the cells. If so, please provide the original unit of the measurement (e. g. $\text{XXX mol O}_2/\text{cell}$). Or it might be better to adjust the initial values at “1” because I could not understand the reason why the authors adjusted the initial values at “about 2.4”.

Answer: We thank you for your valuable comment. We have revised Figure 4 by adjusting the initial values to 1.

Reviewers' comments:

Reviewer #2 (Remarks to the Author):

I thank the authors for clarifying my concerns and for their efforts in answering all the points. They have now included superoxide dismutase activity to assess the in vivo oxidative stress status under salt stress with HL or LL treatment. Their data clearly shows that ki-4N experience more stress in salt plus HL condition, under condition where AstaPs expression is induced.

Reviewer #3 (Remarks to the Author):

I found that the author tried to address my concern by discussing the current dataset. However, I could not be convinced by the author's rebuttal. I attached the files to show which parts were not convincing. Sincerely.

We thank you for the valuable comments. We have revised the manuscripts accordingly.

1. Why the authors used 2 weeks stressed cells for Figure 4c/d, while 1-4 days stressed cells were used for Figure 1c/d/e and Figure 4a? From Figure 1c/d, the alga accumulated clear amounts of AstaP proteins within 2-4 days. Is there any reason to use 2 weeks stressed cells for Figure 4c/d experiments?

Answer: We provided the results from short-term-stressed cells in Fig. 1 because we wanted to show that the AstaP proteins have the ability to respond to the early stage of photooxidative stress. On the contrary, changes in the expression pattern of AstaP-pinks and AstaP-orange2 were observed under long-term stress conditions; the results suggested differences in the functionality of these proteins. Therefore, we presented the data obtained under 2-week stress conditions in Fig. 4. We have revised the relevant sentence to address your concern (line 196).

>I understood the authors answer, but I still have question about this. Because the authors tested the gene expression of *astaP* within 4 “days” in Figure 4a, I could not understand the reason why they showed “2 weeks” treated samples in Figure 4b-d. I think this could be misleading point in Figure 4. The time-scales in the Figure 4 are not consistent.

Moreover, I could not find the “relevant sentence (explanation)” in the revised manuscript.

Answer: We thank you for your valuable comment. As mentioned in our response earlier, we were interested in the early response of the *astaP* genes. Therefore, we have revised the title of Figure4 (line 676). We hope that the revised title addresses your concern.

Regarding your comment “Moreover, I could not find the relevant sentence....,” we apologise for our mistake in indicating the line number; the relevant information is presented in line 199, and not in line 196 as previously indicated.

2. I found that there is only little correlation between gene expression and protein expression of AstaP proteins. Although AstaP proteins were clearly expressed within 2 days of stress treatment (Figure 1c/d/e), its genes seemed to be not much expressed at

that time (Figure 4a). The AstaP genes seemed to be highly expressed after 4 days of stress treatment. I would like to suggest the authors perform qRT-PCR of the genes to show a more clear correlation between gene and protein expression of AstaP.

Answer: In this study, we performed northern blotting to observe the expression of the target genes. We have replaced the image in Fig. 4a with that obtained at a higher exposure time. We hope that you can clearly observe the gene expression, especially the difference between undetectable gene expression under non-stressed condition and induced gene expression under 1-day stress condition. Although qRT-PCR is one of the valuable tools to quantify gene expression, we did not prefer this method because it depends on amplification efficiency of the target gene, and the data may contain unspecific gene amplification from the background genes. Therefore, we used northern blotting to detect the specific expression of the target gene. As you commented, the data of high induction under 4-day stress condition is very interesting. We are thinking that the astaP gene might be responsive to several signals including photooxidative stress. It is well known that the changes in gene expression are frequently not reflected at the protein translation level, and this inconsistency depends on various biochemical regulation mechanisms.

> I strongly disagree with the authors idea about qRT-PCR. First, we can evaluate and compensate the amplification efficiency by testing the primer set used in qRT-PCR. Second, we can confirm the specificity of the gene amplification by sequencing the PCR products after qRT-PCR analysis. Therefore, the reasons raised by the authors are not convincing.

On the other hand, although the authors speculated the reason of an inconsistency between gene expression and protein expression in this rebuttal letter, there is no explanation about this in the revised manuscript. This is not kind enough for the potential readers of the manuscript. In addition, I could not be convinced about the inconsistency between the gene expression and protein expression in Figure 4 by the authors explanation. I am still puzzling about the current Figure 4.

Answer: We thank you for your suggestion and advise. However, we do not have sufficient skills and equipment to perform qRT-PCR, we chose Northern analysis as an accurate method. We believe that this is one of the most accurate methods to evaluate

gene expression profiles. We have revised the title of Figure 4 and added the details of the result (lines 192, 200, 676).

4. What is the unit for “Relative photosynthesis” in Supplementary Figure 4? From the figure legend, I predicted that the authors measured the oxygen evolution activity of the cells. If so, please provide the original unit of the measurement (e. g. XXX mol O₂/cell). Or it might be better to adjust the initial values at “1” because I could not understand the reason why the authors adjusted the initial values at “about 2.4”.

Answer: We thank you for your valuable comment. We have revised Figure 4 by adjusting the initial values to 1.

> Thank you for revising the figure. I thought that the authors have recalculated (probably by dividing the previous valued with 2.4), and the recalculation should not affect the shape of error bars. However, I found that the error bars in the revised Figure seems to be changed from the previous figure. This may be my misunderstanding of the recalculation, but please re-check the dataset.

Answer: We thank you for your valuable comment. During our previous revision of the manuscript, we found some minor errors in original Supplementary Fig. 4, which we fixed. Furthermore, per your suggestion, we adjusted the initial values to 1, and this time we rechecked the data.